# Successful Treatment of Fungal Dermatitis in a Bottlenose Dolphin (*Tursiops truncatus*)

**DOI:** 10.3390/microorganisms13010106

**Published:** 2025-01-07

**Authors:** Takashi Kamio, Honoka Nojo, Rui Kano, Mami Murakami, Yukako Odani, Koji Kanda, Tomoko Mori, Yuichiro Akune, Masanori Kurita, Ayaka Okada, Yasuo Inoshima

**Affiliations:** 1Port of Nagoya Public Aquarium, Nagoya Port Foundation, 1-3 Minato-machi, Minato-ku, Nagoya 455-0033, Japan; 2Laboratory of Food and Environmental Hygiene, Joint Department of Veterinary Medicine, Gifu University, 1-1 Yanagido, Gifu 501-1193, Japan; 3Joint Graduate School of Veterinary Sciences, Gifu University, 1-1 Yanagido, Gifu 501-1193, Japan; 4Teikyo University Institute of Medical Mycology (TIMM), Teikyo University, 359 Otsuka, Hachioji 192-0395, Japan; 5Laboratory of Veterinary Clinical Pathology, Joint Department of Veterinary Medicine, Gifu University, 1-1 Yanagido, Gifu 501-1193, Japan

**Keywords:** debridement, dematiaceous fungi, *Fusarium oxysporum*, pheohyphomycosis, polyaminopropyl biguanide and betaine, terbinafine, *Tursiops truncates*, voriconazole

## Abstract

In recent decades, many fungi have emerged as major causes of disease in marine mammals. This study reports on the detection of filamentous fungi in the subcutaneous tissue and wound surface on the tail fin of a managed bottlenose dolphin (*Tursiops truncatus*) emaciated due to severe digestive problems. Immunosuppression by chronic diseases and starvation decreased resistance against opportunistic infections. Sequencing analysis revealed that the fungi on the wound surface were *Fusarium oxysporum,* and antifungal susceptibility testing was performed. In the subcutaneous tissue, dematiaceous fungi were identified using histopathological examination. Combination antifungal treatment with voriconazole and terbinafine and surgical resection were performed, in addition to daily debridement with polyaminopropyl biguanide (PHMB) and betaine. Hematological examination revealed a reduction in inflammatory markers after antifungal treatment and surgical resection of necrotic tissue on the edge of the tail fin. The co-administration of synergistic agents voriconazole and terbinafine, in conjunction with surgical debridement, successfully eliminated pheohyphomycosis and fusariomycosis in the bottlenose dolphin. Wound healing was achieved using systematic antifungals and daily debridement with PHMB and betaine. This is the first report of successful treatment of pheohyphomycosis and fusariomycosis in a bottlenose dolphin using voriconazole and terbinafine combination therapy and surgical resection.

## 1. Introduction

Deep fungal infections cause extensive destruction of the skin and subcutaneous tissues [1] and can spread via the subcutaneous, disseminated, visceral, and systemic routes [1]. Fungal implantation is the result of trauma in exposed areas, which forms a primary nodular lesion that subsequently grows into a subcutaneous abscess [1]. In recent decades, many fungi have emerged as major causes of disease in marine mammals [2]. Here, we report a case of successful treatment of severe fungal dermatitis in a bottlenose dolphin (*Tursiops truncatus*) with pheophyphomycosis and fusariomycosis. Pheohyphomycosis is an uncommon superficial, subcutaneous, or systemic infection caused by dematiaceous fungi in humans and animals [1]. To date, approximately 70 genera and 150 species of dematiaceous fungi have been described [1]. The growth requirements of the dematiaceous fungus are unique. It thrives at an optimal temperature of 30 °C and grow slowly at 37 °C, requiring a minimum of four weeks for culture [3]. Treatment options for pheohyphomycosis are limited. A combination of surgical excision or debridement, along with the maximum recommended dosage of antifungal drugs, is typically preferred. Voriconazole, when combined with terbinafine, has produced more favorable therapeutic outcomes than when prescribed alone [3]. *Fusarium* spp. is common in marine environments, including aquaria [4]. Previous studies have identified *Fusarium* spp. as an opportunistic fungal pathogen in cetaceans housed in aquaria [4], inducing a range of clinical manifestations [5], including skin lesions [2,5,6,7]. Marine mammals with compromised immune systems, presumably due to stress or illness, are most susceptible [2,7]. Control of the progression of fusariomycosis using a single antifungal agent is challenging because *Fusarium* spp. is also resistant to several antifungal agents [8]. Successful treatment of fusariomycosis using combination therapy comprising voriconazole and terbinafine accompanied by surgical resection of lesions has been reported in humans [8], similar to the treatment for dematiaceous fungi [3]. Additionally, in cetaceans, oral voriconazole and topical treatment with a combination of terbinafine, voriconazole, and dimethylsulfoxide (DMSO) (20 g, 200 mg, and 15 mL, respectively, per application) have reportedly had the most positive effect on healing in a beluga whale (*Delphinapterus leucas*) [6]. Here, we describe a case of a bottlenose dolphin diagnosed with fungal dermatitis caused by dematiaceous fungi and *Fusarium oxysporum*.

## 2. Case Report

A nodule was found on the tail fin of a managed female bottlenose dolphin, estimated to be 26 years old by body length [9] on 22 January 2023 (day 0), at the Port of Nagoya Public Aquarium (Nagoya, Japan). The dolphin had severe digestive problems, characterized by the accumulation of copious amounts of gas in the intestine and frequent egress of undigested fish bones from the anus, which led to adherence and obstruction of stool excretion several times. Comprehensive antibiotic treatment was implemented (Appendix A) with bacterial identification and antibiotic susceptibility testing as described previously [10] because of the animal’s life-threatening situation. The overuse of antibiotics should not be encouraged, especially in light of global alerts for the spread of antimicrobial resistance. However, in this case, the individual was in a critical condition, necessitating the use of multiple classes of antibiotics, covering almost all possible mechanisms of action. The antibiotic dose was based on a previous study of cetaceans [11] and previous prescriptions provided at this facility. The daily diet was limited due to digestive issues; consequently, the dolphin’s weight decreased from an average 265 kg (262–270 kg, from 1 November 2023 to 31 December 2023, day −82 to −22) to 197 kg (minimum, 22 February 2023, day 29), which led to suppression of the dolphin’s immune system.

The bottlenose dolphin was trained to permit blood sampling following the voluntary presentation of a tail fin. Plasma fibrinogen levels were measured using the COAG2NV blood coagulation analysis system (A&T, Tokyo, Japan). The complete blood count was measured using a Celltac α MEK-6558 automatic cell counter (Nihon Kohden, Tokyo, Japan). Leukocyte differentials were determined using cytological examination by counting 150–200 leukocytes in blood smears with May–Gruenwald’s solution (Sigma-Aldrich, Tokyo, Japan) and Gimenez’s solution (Sigma-Aldrich). Each type of leukocyte count (/µL) was calculated from the total leukocyte count and leukocyte differentials. Serum ferrous was measured at a clinical testing company (Nagoya Rinsho, Nagoya, Japan).

Hematological examination revealed elevation in the total leukocyte count (peak: 25,900/µL on day 58, standard range: 5000–9000/µL [12]) and fibrinogen level (peak: 506 mg/dL on day 51, standard range: 170–280 mg/dL [12]), and reduction in the serum ferrous level (peak: 61 on day 49, standard range: 120–340 μg/dL [12]). These findings indicated clinical inflammation in the dolphin (Figure 1). Initially, bacterial infection was considered. Bacterial identification and antibiotic susceptibility testing with the dermatitis samples were conducted as described previously [10]. However, despite comprehensive antibacterial treatment (Figure 1, Appendix A) and minor surgical debridement on days 8, 41, and 51, the nodules progressed to severe dermatitis.

The swabbed dermatitis specimens were cultured on Sabouraud dextrose agar plates on day 8 (BD Japan, Tokyo, Japan). The culture was incubated for 48 h at 37 °C in an IC-450A incubator (AS ONE Corporation, Osaka, Japan), and no fungi were isolated. The plate was placed at room temperature for another 48–72 h and an unidentified filamentous fungus was isolated. Since the unidentified, delay-grown filamentous fungus at room temperature has been observed several times with other specimens, most of which were the result of contamination, it was not included in our routine fungal identification procedures. Thus, identification was not performed on this occasion. The swabbed dermatitis specimens were cultured on day 45, and the same type of unidentified filamentous fungi that were observed on day 8 were found again (Figure 2a). Following staining with lactophenol cotton blue (Muto Pure Chemicals Co., Ltd., Tokyo, Japan), the isolated filamentous fungi showed septate and moderately curved macroconidia, which is characteristic of *Fusarium* spp. [8] (Figure 2b). The fungi were identified as the main cause of the dermatitis. The cultured fungi were sent to the Teikyo University Institute of Medical Mycology for identification and antifungal susceptibility testing.

Following incubation on Sabouraud dextrose agar plates for 5 days, approximately 10 mg of growing mycelia was collected from the clinical isolate and frozen with liquid nitrogen, and genomic DNA was extracted using phenol and ethanol precipitation. The molecular characteristics of the isolates were identified using sequence analysis of the internal transcribed spacer (ITS) region. The ITS region of each isolate was amplified using the universal fungal primers ITS5 (5′-GGAAGTAAAAGTCGTAACAAGC-3′) and ITS4 (5′-TCCTCCGCTTATTGATAGC-3′) [13]. Thirty cycles of polymerase chain reaction (PCR) amplification were performed under the following conditions: denaturation for 30 s at 95 °C, primer annealing for 30 s at 55 °C, and extension for 1 min at 72 °C in a total reaction volume of 30 µL amplification mixture [10 mM Tris-HCl (pH = 8.3), 50 mM KCl, 1.5 mM MgCl_2_, 0.001% gelatin, 200 mM deoxynucleotide triphosphate, 1.0 U Taq polymerase (Takara, Kyoto, Japan), and 50 μM of each primer]. The amplified DNA fragments were electrophoresed on 2% (*w*/*v*) agarose gel in 1× Tris-acetate-ethylenediaminetetraacetic acid buffer and visualized under ethidium bromide staining. For each isolate, a DNA band of approximately 560 base pairs was excised from the gel, purified using the ExoSAP-IT kit (USB Corporation, Cleveland, OH, USA), and sequenced using an ABI PRISM 3130 genetic analyzer (Thermo Fisher Scientific, Inc., Tokyo, Japan). Comparative sequence analyses were performed using the Basic Local Alignment Search Tool (BLAST) on the National Center for Biotechnology Information (NCBI) website (https://blast.ncbi.nlm.nih.gov/Blast.cgi; accessed 14 April 2023). Two fungal specimens isolated from the dermatitis lesions on 11 March 2023 (day 48) and 14 March 2023 (day 51) were identified, and the ITS sequences of the isolates were 100% identical to the *F. oxysporum* ITS region. The cause of the dermatitis was confirmed to be *Fusarium* spp. The analyzed sequences were deposited in GenBank (accession nos. MT447537 and MT530243) and designated as NaTT12-230311 and NaTT12-230314, respectively.

Antifungal susceptibility testing of *F. oxysporum* was performed. In this study, to assess the azole susceptibility of isolate NaTT12-230311, a broth microdilution assay was performed according to the CLSI M38 guidelines [14]. Antifungal susceptibility testing revealed that the minimum inhibitory concentrations (MICs) of voriconazole and posaconazole for NaTT12-230311 were reduced by synergism with terbinafine (Table 1), proving that *F. oxysporum*, NaTT12-230311, is susceptible to the combination of voriconazole and terbinafine.

Combination antifungal therapy with voriconazole (876179, DSEP, Tokyo, Japan) and terbinafine (87629, Sandoz, Tokyo, Japan) [2,18] was initiated on days 48–221 and 51–211, respectively. Due to the dolphin’s life-threatening condition, delaying antifungal therapy by waiting for the results of antifungal susceptibility testing was not feasible. It has been reported that voriconazole and terbinafine showed synergism against *Fusarium* spp. in 5 of 29 cases (17.24%), whereas no reaction was observed in 19 of 29 cases (65.52%), and antagonism was seen in 5 of 29 cases (17.24%) [18]. Combination therapy comprising voriconazole and terbinafine was delivered as soon as *Fusarium* spp. infection was confirmed. Antifungal susceptibility testing results were confirmed on day 67 (Table 1) and proved that combination antifungal therapy had a synergistic effect against the isolated *F. oxysporum*. Plasma voriconazole concentrations were assessed by a clinical testing company (SRL, Inc., Nagoya, Japan). The target plasma voriconazole trough concentration was set at 3.0 mg/L [19]; therefore, the loading dose [4.9 mg/kg per os (P.O.) semel in die (SID) on the first day, and 2.4 mg/kg P.O. bis in die (BID) on the second and third days] and maintenance dose (ranging from 2.4–3.6 mg/kg P.O. BID, every 3–14 days based on plasma concentrations) of voriconazole were prescribed. Terbinafine was given at a dose of 1.9 mg/kg P.O. SID. Topical treatment with a combination of terbinafine, voriconazole, and DMSO [6] was attempted, but was abandoned after a short time because, contrary to the information, DMSO would not remain in contact with the lesion for a long time after submersion of the animal in water. Because adequate wound management in cetaceans is challenging, antibiotics were administered as a prophylactic measure to prevent secondary infections (Appendix A).

To prevent fungal transmission from dermatitis, the tail fin edge was surgically removed on days 64 and 85 (Figure 1). A combination of midazolam 0.048 mg/kg (KS6769, Sandoz; intramuscular [I.M.]) and butorphanol 0.024 mg/kg (VETLI5, Meiji Seika Pharma Co., Ltd., Tokyo, Japan; I.M.) was administered 30 min prior to each surgery to induce sedation [20]. Topical anesthesia was provided using lidocaine 2% (614435304; Sandoz).

Histopathological examination of the excised tail fin edge revealed granulomas intermingled with neutrophils in the subcutaneous tissue (Figure 3a). Brown hyphae were scattered within the granulomas (Figure 3a). The fungi were 1–3 μm thick, branched, segmented, and straight, although some were also curved (Figure 3a: insert). Additionally, the epidermis was ulcerated, and extensive necrotization was evident in the superficial dermis. Based on these findings, the dolphin was diagnosed with a dematiaceous fungal infection. A large number of hyphae were also scattered in the necrotic tissue. Some fungi were brownish, whereas others had a bluish coloration, showing marked curvature, branching, and septation (Figure 3b). Periodic acid–Schiff staining showed that the hyphae had a remarkable red-purple color (Figure 3b, insert). These findings revealed that the dematiaceous fungi were located in the granulomatous inflammation area with a host response and that *F. oxysporum* was located on the superficial necrotic area without a host response. These findings suggested that pheohyphomycosis with dematiaceous fungi was the primary infection, and fusariomycosis with *F. oxysporum* was the secondary infection.

Antifungal susceptibility testing for dematiaceous fungi could not be performed because the fungi were not cultured. However, combination therapy (voriconazole and terbinafine) against *F. oxysporum* and the surgical approach were continued because this protocol has also been recommended for the treatment of dematiaceous fungal infection [3,21]. Necrotic tissue was found on the dorsal side of the tail fin. The extent of tissue removal was minimized by selectively removing the necrotized tissue from the dorsal side while preserving the non-damaged tissue on the ventral side (Figure 4). The healing process was accelerated following granulomatous tissue removal. Among the bottlenose dolphins in the same aquarium, excluding the affected dolphins, none were infected with dematiaceous fungi or *F. oxysporum*. The fungal infection on the tail fin, which would normally not be problematic, progressed to a severe condition, probably because the weight loss due to gastrointestinal disease resulted in a decrease in the immunity of the dolphin.

During antifungal treatment, daily debridement was implemented (days 51–174) (Figure 4). Starting on the day of the second surgical approach (day 85), the wound was treated with polyaminopropyl biguanide (PHMB) (0.1%) and betaine (0.1%) wound gel (Prontosan, B. Braun Medical AG, Sempach, Switzerland) for an average of 12.1 min/day (days 85–102) (Figure 4). The combination of PHMB and betaine could more effectively penetrate resilient wound coatings, lift debris, and clear biofilms from the wound [22]. The removal of the biofilm is anticipated to contribute to the inhibition of fungal growth. Daily debridement was discontinued on day 174 because the wound had almost completely resolved (Figure 4). Combined treatment with voriconazole and terbinafine was effective against both the dematiaceous fungus and *Fusarium* spp., preventing further spread of the lesion and contributing to the improvement of inflammatory markers in hematological examinations.

## 3. Discussion

Dermatitis with *F. oxysporum* has been reported to be locally invasive or a manifestation of systemic disease [3]. *F. oxysporum* infection in the brain has been reported in bottlenose dolphins as a cause of death [2]. In cetaceans, it has been reported that chronic diseases and starvation are associated with a loss of functional lymphoid cells and decreased resistance against opportunistic infections [23], including dematiaceous fungi and *Fusarium* spp. [2]. The reasons for severe digestive problems in this dolphin were not identified. However, weight gain was achieved (253 kg on day 215, average, 0.25 kg weight/day) by increasing daily food intake by feeding finely chopped fish including/excluding bones, without adherence and obstruction of stool excretion by undigested bones. Improvement of the immune status by the weight gain should have contributed to the eradication of the fungal infection.

Although still unproven, the combination of azoles and terbinafine shows promise for the treatment of patients with *Aspergillus* spp., Mucorales, *Fusarium* spp., and resistant *Candida* spp. infections [2]. Successful treatment with a combination of oral voriconazole and terbinafine, similar to this study, has been reported in a bottlenose dolphin with respiratory infection caused by *Coccidioides immitis* [2]. The doses for coccidioidomycosis were voriconazole 3.5–5.8 mg/kg PO once a week with terbinafine 5.8 mg/kg PO SID (401 days) [2]; diagnostic methods were histopathology, culture, PCR, and hematological examination [2]. Coccidioidomycosis is a systemic fungal infection endemic in Southwestern United States, Mexico, and Central and South America [24]. *C. immitis*, one of the causal agents that appears to be highly susceptible to most antifungal agents, including voriconazole [24,25] and terbinafine [25], is one of the true pathogens that invade tissues of an immunocompetent host [3]. Compared to *C. immitis*, the synergistic approach between voriconazole and terbinafine was more important for the dematiaceous fungi and *F. oxysporum*, because these fungi have high resistance to multiple antifungals, including voriconazole or terbinafine [3,8]. To the best of our knowledge, this study is the first report of successful treatment of pheohyphomycosis and fusariomycosis in a bottlenose dolphin using a combination of voriconazole and terbinafine therapy and surgical resection.

There was a possibility of no synergy between azoles and terbinafine against both isolated dematiaceous fungi [21] and *F. oxysporum* [18]. When the infecting fungi show high resistance or antagonism to azoles and terbinafine, minocycline, a broad-spectrum semisynthetic tetracycline with anti-inflammatory, anti-apoptotic, immune-modulatory, neuroprotective, and antiviral properties, has shown potential in combating highly resistant fungi due to its synergistic effects with azoles, such as itraconazole, voriconazole, and posaconazole, including against azole-resistant *A. fumigatus* and *Fusarium* spp. [26]. Minocycline combined with azoles can enhance the antifungal susceptibility of azoles against pathogenic fungi and have the potential to overcome azole resistance issues [26]. Synergistic medical approaches have the potential to treat highly resistant fungal infections successfully.

In conclusion, the administration of synergistic agents voriconazole and terbinafine, in conjunction with surgical debridement, proved successful in treating pheohyphomycosis and fusariomycosis in a bottlenose dolphin. Infected necrotic tissue with no chance of recovery should be excised as part of the treatment. Although the dolphin experienced a slight reduction in jumping height, by preserving as much of the tail fin as possible, the extent of the resection was successfully limited to a range that did not affect normal swimming. Wound healing was achieved using systematic antifungals and daily debridement with PHMB and betaine.

## Figures and Tables

**Figure 1 microorganisms-13-00106-f001:**
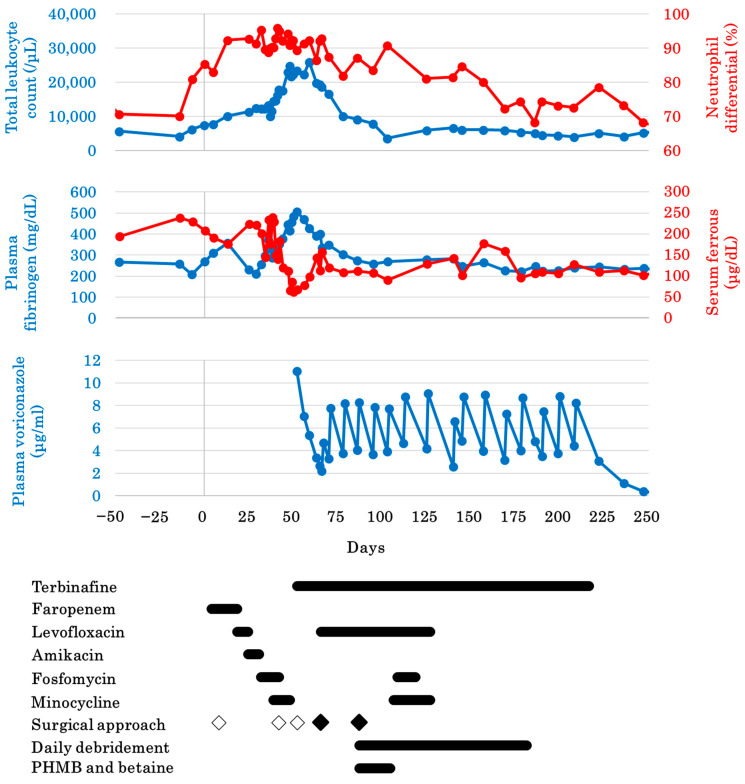
Hematological progress of the bottlenose dolphin (*Tursiops truncatus*) treated for fungal dermatitis. On day 0, a nodule was found on the tail fin of the managed female bottlenose dolphin. White diamond, minor surgical debridement (days 8, 41, and 51); black diamond, surgical resection (days 64 and 85). Detailed doses and duration of medicines are described in Appendix A.

**Figure 2 microorganisms-13-00106-f002:**
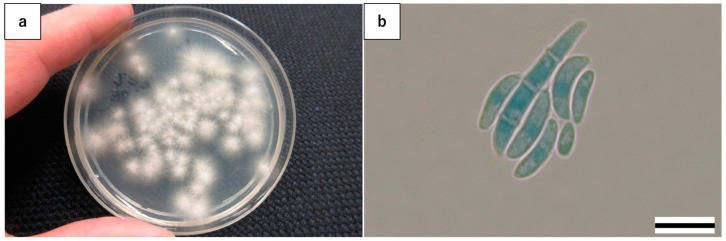
Dermatitis on the tail fin of a bottlenose dolphin (*Tursiops truncatus*). (**a**) Isolated filamentous fungi on a Sabouraud dextrose agar plate. (**b**) Isolated filamentous fungi with septate, moderately curved macroconidia, characteristic of *Fusarium* spp., viewed with lactophenol cotton blue staining (Bar = 10 μm).

**Figure 3 microorganisms-13-00106-f003:**
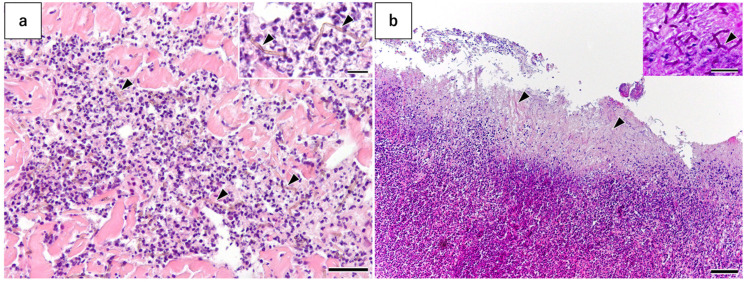
Histopathological examination of the excised tail fin. (**a**) In the subcutaneous tissue, granulomas mixed with neutrophils were observed. Brown fungal hyphae (arrowheads) are scattered within the granuloma. The fungi were 1–3 μm thick, branched, segmented, and straight, although some were curved (insert) (bar = 25 μm; bar (insert) = 10 μm). (**b**) The epidermis was ulcerated and defective. A wide area of necrosis was seen in the superficial layer of the dermis, with many hyphae observed in the necrotic tissue. Fungal growth in necrotic tissue showed a slightly bluish color, a slightly more robust curvature with an average diameter of 2–5 μm, and bifurcations and septa (arrowheads). Brown fungi were also identified in the necrotic tissue. Periodic acid–Schiff staining revealed that the hyphae were markedly red-purple (insert; arrowhead) (bar = 250 μm; bar (insert) = 10 μm).

**Figure 4 microorganisms-13-00106-f004:**
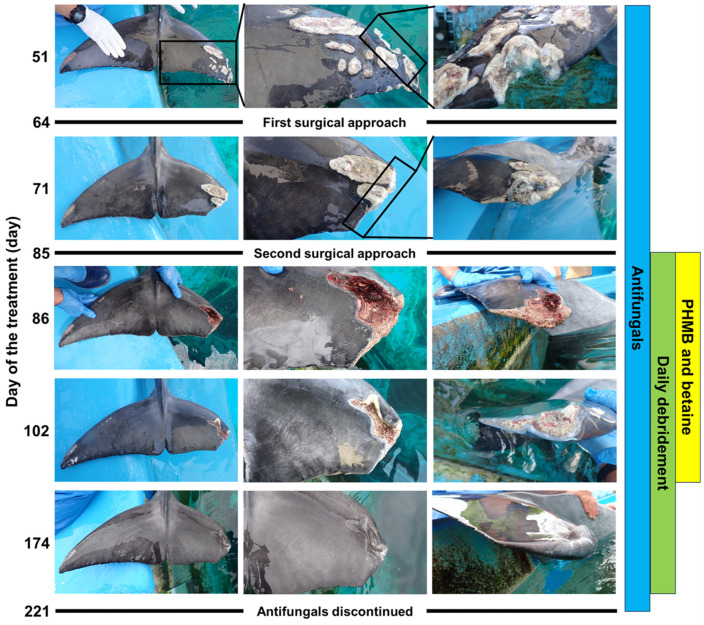
Wound healing progress. Left panels: the tail fin from the dorsal side. Center panels: wound on the tail fin from the dorsal side. Right panels: cross section of the wound. PHMB, polyaminopropyl biguanide.

**Table 1 microorganisms-13-00106-t001:** Minimum inhibitory concentration (MIC) of antifungal agents for isolated *Fusarium oxysporum* (NaTT12-230311) from dermatitis in a bottlenose dolphin (*Tursiops truncates*).

Antifungal Agent	MIC (mg/L)	Susceptibility	Interpretation			Reference
			Susceptible	Intermediate	Resistance	
Single agent						
ITCZ	>32	Resistance	<2	2	>2	[15]
VRCZ	2	Intermediate	<2	2	>2	[15]
PSCZ	1	Resistance	<0.25	0.5	>0.5	[15]
FLCZ	>32	Resistance	≤8	16–32	>64	[16]
RVCZ	0.5	Susceptible	≤1	Not specified	≥2	[16]
AMHB	4	Resistance	≤1	Not specified	>2	[17]
Combination with terbinafine						
VRCZ	1	Susceptible	<2	2	>2	[15]
PSCZ	0.5	Intermediate	<0.25	0.5	>0.5	[15]
RVCZ	0.5	Susceptible	≤1	Not specified	≥2	[16]
AMHB	2	Resistance	≤1	Not specified	>2	[17]

ITCZ, itraconazole; VRCZ, voriconazole; PSCZ, posaconazole; FLCZ, fluconazole; RVCZ, ravuconazole; AMHB, amphotericin B.

## Data Availability

The data presented in this study are available within the article and in Appendix A.

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
