# Peer review of "Successful Treatment of Fungal Dermatitis in a Bottlenose Dolphin (Tursiops truncatus)"

_microorganisms, 2025, doi:10.3390/microorganisms13010106_

Round 1

Reviewer 1 Report

Comments and Suggestions for Authors

The present manuscript describes a case report of mycotic infection by opportunistic fungi in a bottlenose dolphin.

The report describes a weird management of the case, and some aspects are difficult to interpretate:

61-63: Fusarium spp. are common in marine environment, and have been reported several times as pathogen for marine species, especially in debilitated/immunocompromised individuals, so they should always be considered in differential diagnosis, primarily because of the difficulty to treat them once the infection is established: why did the authors not request an identification of the “contaminant” filamentous fungi?

64-65: what was the hypothesized connection between a gastrointestinal infection and a cutaneous nodule on caudal flipper?

79: table S1 reports an impressive list of antimicrobial drugs… is this a standard protocol? Is the posology chosen according to scientific literature on dolphins? Do the authors perform any laboratory test (isolation, sensitivity, MIC) on any bioptic material from the nodule? There are many different classes of antimicrobials in the list, covering almost any possible mechanism of action, just as if the aim was to intercept an unknown enemy. In the age of global alert for the spread of antimicrobial resistance, it is a quite irresponsible use of drugs, especially those listed as essential life-saving drugs for humans.

Even the list of drugs in table S2 (after fungal diagnosis) has the same chaotic association of antimicrobials, and once again the authors report no laboratory testing to justify this choice.

Jump to 214: if a bacterial infection was the prior diagnosis, and antibacterial treatment the priority, why none of the essentials in establishing an antimicrobial therapeutic protocol has been applied?

138: in previous paragraphs, it seemed that laboratory testing had indicated the association terbinafine-voriconazole as active on the isolate of Fusarium, and on this basis the treatment was decided, but here the authors report that the prescription was decided “prior to fungal identification…. and antifungal susceptibility testing results were confirmed”. Please, add more explanations

187-188: fungi, and especially ubiquitarians like Fusarium, are generally opportunistic pathogens, so an “occasion” is required to establish an infection. In the present case, probably the non-specified gastrointestinal infection had debilitated the dolphin, and any banal cutaneous lesion may have served as an entrance for the infection, while immune response was low.

220-221: this is a great truth, antimicrobials should never be used injudiciously, and the need for therapy should be established on a laboratory basis, as the clinical failure is always possible because of the spread of resistances, even with the “best practice” procedures.

Overall, the useful part of the manuscript is the report of the possibility of a therapy for a fastidious fungal species, but this could be limited to a short communication or a letter, as the whole management of the clinical case looks like “all that you shouldn’t do”.

Author Response

Responses to Reviewer 1’s comments

First of all, thank you very much for your valuable comments and suggestions. We have revised our manuscript according to your comments and suggestions. The text changes in the manuscript are highlighted in yellow color. Our replies to your comments are listed below.

Reviewer 1

Comments to the Author

The present manuscript describes a case report of mycotic infection by opportunistic fungi in a bottlenose dolphin.

The report describes a weird management of the case, and some aspects are difficult to interpretate:

61-63: Fusarium spp. are common in marine environment, and have been reported several times as pathogen for marine species, especially in debilitated/immunocompromised individuals, so they should always be considered in differential diagnosis, primarily because of the difficulty to treat them once the infection is established:

Our response

According to your suggestion, we have added the information on Fusarium infection in dolphins in the introduction (L47-51).

Comments to the Author

why did the authors not request an identification of the “contaminant” filamentous fungi?

Our response

According to your comments, we have revised the text (L104-107).

Unfortunately, Fusarium spp. was not included in our routine fungal identification procedures. Since the delay-grown unidentified filamentous fungus in room temperature has been observed several times with other specimens and most of them came from contamination, it was not included in our routine fungal identification procedures. For this reason, identification was not conducted on this occasion at first. As you pointed out, in the future, we plan to include fungal species identification. We kindly ask for your understanding regarding the circumstances in this matter.

Comments to the Author

64-65: what was the hypothesized connection between a gastrointestinal infection and a cutaneous nodule on caudal flipper?

Our response

According to your comments, we have revised the text (L65–75, L217–220).

Comments to the Author

79: table S1 reports an impressive list of antimicrobial drugs… is this a standard protocol? Is the posology chosen according to scientific literature on dolphins?

Our response

According to your comments, we have revised the text (L70–L72).

Comments to the Author

Do the authors perform any laboratory test (isolation, sensitivity, MIC) on any bioptic material from the nodule?

Our response

Yes, we did. However, to focus the report to the fungal dermatitis, detailed information was not described.

According to your comments, we have revised the text (L90–91).

Comments to the Author

There are many different classes of antimicrobials in the list, covering almost any possible mechanism of action, just as if the aim was to intercept an unknown enemy. In the age of global alert for the spread of antimicrobial resistance, it is a quite irresponsible use of drugs, especially those listed as essential life-saving drugs for humans.

Even the list of drugs in table S2 (after fungal diagnosis) has the same chaotic association of antimicrobials, and once again the authors report no laboratory testing to justify this choice.

Our response

We are fully aware of the importance of avoiding the overuse of antibiotics. In this case, the individual was in such a critical condition that we had no choice but to prescribe a broad spectrum of antibiotics (L65–75).

In the case of dolphins, after partial removal of the tail fin, adequate wound management is challenging. Therefore, we continued antibiotic administration as a preventive measure against secondary infection (L172–177).

Comments to the Author

Jump to 214: if a bacterial infection was the prior diagnosis, and antibacterial treatment the priority, why none of the essentials in establishing an antimicrobial therapeutic protocol has been applied?

Our response

During the debridement of the dermatitis on the tail fin, bacterial identification and antibiotics susceptibility testing were conducted. However, these details were not included as they fall outside the scope of this case.

According to your comments, we have revised the text (L90–91).

Comments to the Author

138: in previous paragraphs, it seemed that laboratory testing had indicated the association terbinafine-voriconazole as active on the isolate of Fusarium, and on this basis the treatment was decided, but here the authors report that the prescription was decided “prior to fungal identification…. and antifungal susceptibility testing results were confirmed”. Please, add more explanations

Our response

We have corrected the text because it was not ‘identification’ but ‘antifungal susceptibility testing’.

This individual was in a critical condition. Once identified as Fusarium, we anticipated limited efficacy with voriconazole alone. Unable to wait for susceptibility testing, we initiated treatment with voriconazole and terbinafine, based on the previous reports.

  • Inano et al. “Combination therapy of voriconazole and terbinafine for disseminated fusariosis: case report and literature review.” Journal of Infection and Chemotherapy 19, 1173–1180, 2013.
  • Córdoba et al. “In vitro interactions of antifungal agents against clinical isolates of Fusarium spp.” International Journal of Antimicrobial Agents 31: 171–174, 2008.

We have corrected the text (L158–166).

Comments to the Author

187-188: fungi, and especially ubiquitarians like Fusarium, are generally opportunistic pathogens, so an “occasion” is required to establish an infection. In the present case, probably the non-specified gastrointestinal infection had debilitated the dolphin, and any banal cutaneous lesion may have served as an entrance for the infection, while immune response was low.

Our response

According to your valuable comments, we have revised the text (L47–51, L217–220).

Comments to the Author

220-221: this is a great truth, antimicrobials should never be used injudiciously, and the need for therapy should be established on a laboratory basis, as the clinical failure is always possible because of the spread of resistances, even with the “best practice” procedures.

Our response

As you pointed out, since it is a well-known fact, we have revised the text (L251).

Comments to the Author

Overall, the useful part of the manuscript is the report of the possibility of a therapy for a fastidious fungal species, but this could be limited to a short communication or a letter, as the whole management of the clinical case looks like “all that you shouldn’t do”.

Our response

As you commented, our manuscript has been submitted as a case report.

Reviewer 2 Report

Comments and Suggestions for Authors

The manuscript is devoted to the analysis of the antibiotics combination activity against dermatitis caused by fungi of the genus Fusarium. The study is well-planned and logically executed. But I have a number of comments.

Figure 1. is difficult to interpret. The authors need to either describe it in detail in the text or, preferably, present the data in a more understandable form (perhaps as a table).

The introduction needs supplementing with the information on the methods of combating Fusarium spp, using the recent references.

The first two paragraphs of the discussion are more about the results. The rest of it is not a proper discussion. I suppose the authors should supplement the part with the  data obtained by other researchers. And also the authors should emphasize the novelty of the study. If the authors postulate that it is the first use an antifungal combination on this object. But have there been any cases of applying similar combination using other animals, or using other dermatitis caused by other fungi?

Author Response

Responses to Reviewer 2’s comments

First of all, thank you very much for your valuable comments and suggestions. We have revised our manuscript according to your comments and suggestions. The text changes in the manuscript are highlighted in yellow color. Our replies to your comments are listed below.

Reviewer 2

Comments to the Author

The manuscript is devoted to the analysis of the antibiotics combination activity against dermatitis caused by fungi of the genus Fusarium. The study is well-planned and logically executed. But I have a number of comments.

Figure 1. is difficult to interpret. The authors need to either describe it in detail in the text or, preferably, present the data in a more understandable form (perhaps as a table).

Our response

To meet with your comments, we revised the Fig. 1 to be much clearer visually.

As you suggested, we have added a detailed description in the text (L97–99). In addition, detailed doses and duration of medicines were described in supplementary Tables S1 and S2.

The graph style was followed the Figure 1 in the previous report.

Ohno et al. "Leukopenia induced by micafungin in a bottlenose dolphin (Tursiops truncatus): a case report." Journal of Veterinary Medical Science 81: 449-453, 2019.

Comments to the Author

The introduction needs supplementing with the information on the methods of combating Fusarium spp, using the recent references.

Our response

As you suggested, we have added information in the text (L56–59).

We have added information in the text about the reason why we did not choose the method which was recommended in a beluga whale (L172–177).

Comments to the Author

The first two paragraphs of the discussion are more about the results. The rest of it is not a proper discussion. I suppose the authors should supplement the part with the data obtained by other researchers. And also the authors should emphasize the novelty of the study. If the authors postulate that it is the first use an antifungal combination on this object. But have there been any cases of applying similar combination using other animals, or using other dermatitis caused by other fungi?

Our response

According to your suggestion, we have revised the discussion.

Round 2

Reviewer 1 Report

Comments and Suggestions for Authors

This is a revised version of the manuscript.

The authors have addressed all the observations, so many questionable aspects of the 1st draft have been clarified.

For a more comprehensive description of the case report, it could be useful to add more information about nature and output of the primary disease (the intestinal infection), to parallel it with the improvement of the immune status and the healing of the fungal infection, since the dolphin's immune response certainly contributed to its eradication (apart from drugs), as Fusarium is always an opportunistic pathogen. Considering the paucity of available literature on cetacean diseases and their treatment, any information can be precious.

Author Response

Responses to Reviewer 1’s comments

First of all, thank you very much for your valuable comments and suggestions. We have revised our manuscript according to your comments and suggestions. The text changes in the manuscript are highlighted in yellow color. Our replies to your comments are listed below.

Reviewer 1

Comments to the Author

The authors have addressed all the observations, so many questionable aspects of the 1st draft have been clarified.

For a more comprehensive description of the case report, it could be useful to add more information about nature and output of the primary disease (the intestinal infection), to parallel it with the improvement of the immune status and the healing of the fungal infection, since the dolphin's immune response certainly contributed to its eradication (apart from drugs), as Fusarium is always an opportunistic pathogen. Considering the paucity of available literature on cetacean diseases and their treatment, any information can be precious.

Our response

As you suggested, we have added the information with the data obtained by other researchers (L244–282).

Added references

  • Beineke, A.; Siebert, U.; Wohlsein, P.; Baumgärtner, W. Immunology of whales and dolphins. Immunol. Immunopathol. 2010, 133, 81–94.
  • Ramani, R.; Chaturvedi, V. Antifungal susceptibility profiles of Coccidioides immitis and Coccidioides posadasii from endemic and non-endemic areas. Mycopathologia. 2007, 163, 315–319.
  • Tintelnot, K.; De Hoog, G.S.; Antweiler, E.; Losert, H.; Seibold, M.; Brandt, M.A.; Gerrits Van Den Endo, A.H.G.; Fisher, M.C. Taxonomic and diagnostic markers for identification of Coccidioides immitis and Coccidioides posadasii. Mycol. 2007, 45, 385–393.
  • Gao, L.; Sun, Y.; Yuan, M.; Li, M.; Zeng, T. In vitro and in vivo study on the synergistic effect of minocycline and azoles against pathogenic fungi. Agents Chemother. 2020, 64, e00290-20.

Reviewer 2 Report

Comments and Suggestions for Authors

I thank the authors for refining the article. However, I still think the discussion needs to be improved. I leave the last remark.

The first two paragraphs of the discussion are more about the results. The rest of it is not a proper discussion. I suppose the authors should supplement the part with the data obtained by other researchers. And also the authors should emphasize the novelty of the study. If the authors postulate that it is the first use an antifungal combination on this object. But have there been any cases of applying similar combination using other animals, or using other dermatitis caused by other fungi?

Author Response

Responses to Reviewer 2’s comments

First of all, thank you very much for your valuable comments and suggestions. We have revised our manuscript according to your comments and suggestions. The text changes in the manuscript are highlighted in yellow color. Our replies to your comments are listed below.

Reviewer 2

Comments to the Author

I thank the authors for refining the article. However, I still think the discussion needs to be improved. I leave the last remark.

The first two paragraphs of the discussion are more about the results.

Our response

As you suggested, we have moved the two paragraphs to the section of introduction and case report in the text (L44–46, L235–238).

Comments to the Author

The rest of it is not a proper discussion. I suppose the authors should supplement the part with the data obtained by other researchers.

Our response

As you suggested, we have added the information with the data obtained by other researchers (L244–282).

Added references

  • Beineke, A.; Siebert, U.; Wohlsein, P.; Baumgärtner, W. Immunology of whales and dolphins. Immunol. Immunopathol. 2010, 133, 81–94.
  • Ramani, R.; Chaturvedi, V. Antifungal susceptibility profiles of Coccidioides immitis and Coccidioides posadasii from endemic and non-endemic areas. Mycopathologia. 2007, 163, 315–319.
  • Tintelnot, K.; De Hoog, G.S.; Antweiler, E.; Losert, H.; Seibold, M.; Brandt, M.A.; Gerrits Van Den Endo, A.H.G.; Fisher, M.C. Taxonomic and diagnostic markers for identification of Coccidioides immitis and Coccidioides posadasii. Mycol. 2007, 45, 385–393.
  • Gao, L.; Sun, Y.; Yuan, M.; Li, M.; Zeng, T. In vitro and in vivo study on the synergistic effect of minocycline and azoles against pathogenic fungi. Agents Chemother. 2020, 64, e00290-20.

Comments to the Author

And also the authors should emphasize the novelty of the study. If the authors postulate that it is the first use an antifungal combination on this object.

Our response

According to your suggestion, we have revised the text (L28–30, L269–271).

Comments to the Author

But have there been any cases of applying similar combination using other animals, or using other dermatitis caused by other fungi?

Our response

According to your suggestion, we have revised the text (L257–269).